# ATP13A2 Regulates Cellular α-Synuclein Multimerization, Membrane Association, and Externalization

**DOI:** 10.3390/ijms22052689

**Published:** 2021-03-07

**Authors:** Jianmin Si, Chris Van den Haute, Evy Lobbestael, Shaun Martin, Sarah van Veen, Peter Vangheluwe, Veerle Baekelandt

**Affiliations:** 1Laboratory for Neurobiology and Gene Therapy, Department of Neurosciences, Leuven Brain Institute, KU Leuven, Herestraat 49, Bus 1023, 3000 Leuven, Belgium; jianmin.si@kuleuven.be (J.S.); chris.vandenhaute@kuleuven.be (C.V.d.H.); evy.lobbestael@kuleuven.be (E.L.); 2Leuven Viral Vector Core, Division Molecular Medicine, KU Leuven, Herestraat 49, Bus 1023, 3000 Leuven, Belgium; 3Laboratory of Cellular Transport Systems, Department of Cellular and Molecular Medicine, KU Leuven, Herestraat 49, Bus 802, 3000 Leuven, Belgium; familyvanmar10@gmail.com (S.M.); sarah.vanveen@kuleuven.be (S.v.V.); peter.vangheluwe@kuleuven.be (P.V.)

**Keywords:** ATP13A2, α-synuclein, α-synuclein multimerization, spermine, Parkinson’s disease

## Abstract

ATP13A2, a late endo-/lysosomal polyamine transporter, is implicated in a variety of neurodegenerative diseases, including Parkinson’s disease and Kufor–Rakeb syndrome, an early-onset atypical form of parkinsonism. Loss-of-function mutations in ATP13A2 result in lysosomal deficiency as a consequence of impaired lysosomal export of the polyamines spermine/spermidine. Furthermore, accumulating evidence suggests the involvement of ATP13A2 in regulating the fate of α-synuclein, such as cytoplasmic accumulation and external release. However, no consensus has yet been reached on the mechanisms underlying these effects. Here, we aimed to gain more insight into how ATP13A2 is linked to α-synuclein biology in cell models with modified ATP13A2 activity. We found that loss of ATP13A2 impairs lysosomal membrane integrity and induces α-synuclein multimerization at the membrane, which is enhanced in conditions of oxidative stress or exposure to spermine. In contrast, overexpression of ATP13A2 wildtype (WT) had a protective effect on α-synuclein multimerization, which corresponded with reduced αsyn membrane association and stimulation of the ubiquitin-proteasome system. We also found that ATP13A2 promoted the secretion of α-synuclein through nanovesicles. Interestingly, the catalytically inactive ATP13A2 D508N mutant also affected polyubiquitination and externalization of α-synuclein multimers, suggesting a regulatory function independent of the ATPase and transport activity. In conclusion, our study demonstrates the impact of ATP13A2 on α-synuclein multimerization via polyamine transport dependent and independent functions.

## 1. Introduction

Parkinson’s disease (PD) is the most common neurodegenerative movement disorder, with prominent loss of dopaminergic neurons in the substantia nigra pars compacta (SNpc) and the presence of proteinaceous inclusions termed Lewy bodies [1]. Dopaminergic neuron loss results in the deficiency of dopamine within the basal ganglia and leads to motor dysfunction, such as resting tremor, rigidity, bradykinesia, and postural instability [2]. The main component of the Lewy bodies is α-synuclein (αsyn) [3,4]. Both missense mutations (A53T/E, E46K, A30P, H50Q, G51D) and duplication or triplication of the *SNCA* gene (encoding αsyn) can cause autosomal dominant PD [5]. In addition, mounting evidence claims that the disruption of various cellular events is mediated by αsyn-induced neurotoxicity and suggests that αsyn acts as a central player in the pathogenesis of PD [6]. αsyn is widely expressed in the central nervous system and is mainly localized in the presynaptic terminals, where it is found both cytosolic and membrane-bound [7,8]. Due to its association with the synaptic membrane to promote membrane curvature, αsyn is strongly suggested as a chaperone protein that regulates the neurotransmitter release [9]. On the other hand, membrane-bound αsyn is reported to promote the formation of intracellular aggregates via recruitment of the cytosolic form [10]. Membrane vesicles and lipids have also been reported to accelerate αsyn fibrillization by increasing the aggregation kinetics [11]. Additional factors such as oxidation are well known to promote αsyn aggregation. Indeed, we previously showed that incubation of neuronal cells with hydrogen peroxidase (H_2_O_2_) and ferrous chloride (FeCl_2_) induces αsyn aggregation [12]. Moreover, imbalance of αsyn synthesis and a progressive decline in proteolytic efficiency can result in abnormal levels of αsyn, which will favor the formation of multimers and oligomers, eventually accumulating into toxic fibrillar species [13].

ATP13A2 is a P5B-type ATPase that is highly expressed in the human brain with highest expression in the SNpc [14], where it localizes to lysosomes and late endosomes [15]. Different mutations in the ATP13A2 gene have been shown to be involved in different neurodegenerative diseases including Kufor–Rakeb syndrome, early-onset PD, hereditary spastic paraplegia, neuroid ceroid lipofuscinosis, and amyotrophic lateral sclerosis [16,17,18,19,20]. To date, pathogenic ATP13A2 mutations have been identified that cause a variety of functional deficits, such as subcellular mislocalization, decreased protein half-life, and the loss of protein function [20,21,22]. Deficiency of ATP13A2 or loss-of-function mutations result in various cellular deficits such as lysosomal impairment, mitochondrial dysfunction, and dyshomeostasis of heavy metals [21,22,23,24,25]. Of interest, αsyn and ATP13A2 seem to be members of the same intracellular network. Both deficiency of ATP13A2 and familial ATP13A2 mutations have been reported to directly affect the fate of αsyn, such as intracellular accumulation and external release [15,21,26,27]. In addition, a link between ATP13A2 and αsyn in the context of PD pathology was supported by the finding that αsyn overexpression-induced toxicity in yeast could be rescued by co-expression of the yeast ATP13A2 ortholog YPK9, and dopaminergic neuron loss in *Caenorhabditis elegans* was antagonized by ATP13A2 [28]. Moreover, αsyn-induced motor deficits were exacerbated in ATP13A2-deficient mice compared to wildtype (WT) mice [29]. On the other hand, no αsyn abnormalities could be observed in ATP13A2-null mice [30] and AAV-mediated overexpression of human ATP13A2 was not protective in an αsyn-based rat model of PD [31]. Taken together, the precise mechanism mediating the crosstalk between ATP13A2 and αsyn remains poorly understood and requires further research.

We recently reported that ATP13A2 is a lysosomal polyamine exporter with the highest affinity for spermine (SPM) [32]. Polyamines (putrescine, spermidine, and SPM) are involved in many cellular processes, including cell proliferation and differentiation, modulation of ion channels and receptors, and regulation of gene transcription and translation [33]. SPM is an antioxidant under physiological conditions [34], and we recently showed that ATP13A2-mediated SPM transport counteracts mitochondrial oxidative stress [35]. However, at high concentration, SPM exerts toxic effects, possibly due to lysosomal accumulation and lysosomal-dependent cell death [32]. Interestingly, a direct link between spermidine or SPM and αsyn aggregation was previously reported, given that polyamines can alter the αsyn conformation, making it more prone to aggregation [36,37]. ATP13A2 deficiency sensitizes cells to SPM toxicity, due to impaired lysosomal activity and loss of lysosomal integrity [32]. Since multiple studies demonstrated a relationship between lysosomal dysfunction and αsyn aggregation [38], in this study, we wanted to study the effect of ATP13A2 on the pathophysiology of αsyn in detail, taking into account these recent findings. We compared basal and stress conditions induced by oxidative stress or SPM and investigated the involvement of ATP13A2 in αsyn membrane association and multimerization. In addition, we studied the proteolytic degradation system and exocytosis to determine whether ATP13A2 contributes to the intracellular removal of αsyn. We show that knockdown (KD) of ATP13A2 impairs lysosomal membrane integrity, together with increased αsyn membrane association and αsyn multimerization. Conversely, ATP13A2 WT plays a protective role against αsyn multimerization by maintaining the integrity of the lysosomal membrane and by inhibiting αsyn membrane association and multimerization. We also found that it regulates the ubiquitin-proteasome system (UPS) and nanovesicle-based external secretion to remove αsyn multimers.

## 2. Results

### 2.1. ATP13A2 KD Results in Upregulation of αsyn Multimers

Deficiency of ATP13A2 and familial ATP13A2 mutations have been reported to induce αsyn aggregation [21,26,28,39]. Here, we aimed to explore effect of ATP13A2 on αsyn multimerization. We generated human neuroblastoma SH-SY5Y cells with stable expression of human αsyn WT (SH-SY5Y-αsyn) combined with microRNA-based short hairpin (sh)-mediated ATP13A2 KD, overexpression of ATP13A2 WT, or overexpression of the ATPase-deficient mutant ATP13A2 D508N (DN). Cell lines expressing sh-firefly luciferase (fLuc) or fLuc were used as a negative control. Comparable overexpression of ATP13A2 WT and DN was confirmed by immunoblotting (Figure A1a, Appendix A). To assess the assembly states of αsyn under native conditions, we performed intact cell cross-linking with disuccinimidyl glutarate (DSG), as described before [40]. The reducible crosslinker dithiobis succinimidyl propionate (DSP) was used as a control, given that its interaction with αsyn can be cleaved by β-mercaptoethanol. Treatment with 0.5 µM DSG resulted in the detection of multiple αsyn multimers. To ensure physiological levels of cross-linking, we checked DJ-1 dimerization as a positive control and the absence of significant cross-linking of tubulin as a negative control (Figure A1b,c, Appendix A). Comparison of the different cell lines revealed that overexpression of ATP13A2 WT or DN did not significantly affect αsyn multimerization under steady-state conditions. However, ATP13A2 KD induced an increase in αsyn multimers (Figure 1a,b). No difference could be detected in αsyn multimers in control sh-fLuc or fLuc cells (Figure A1d, Appendix A). These findings indicate that deficiency in ATP13A2 increases the presence of αsyn multimers under normal conditions.

### 2.2. ATP13A2 Regulates Oxidative Stress-Induced αsyn Multimerization

Next, we examined the effect of ATP13A2 on αsyn under cellular stress. We previously showed that oxidative Fenton stress caused by FeCl_2_ and H_2_O_2_ induces αsyn aggregation [12]. Here, we investigated the impact of Fenton stress on αsyn multimerization in cells overexpressing ATP13A2 KD, ATP13A2 WT, or ATP13A2 DN. We confirmed increased αsyn multimerization under these conditions using cross-linking (Figure 2a,b or Figure A2a, Appendix A). The oxidative stress-induced αsyn multimerization was prevented in conditions of ATP13A2 WT overexpression, but not by the ATPase inactive mutant ATP13A2 DN (Figure 2c,d or Figure A2b, Appendix A).

### 2.3. ATP13A2 Regulates αsyn Membrane Association and Multimerization

We further investigated whether ATP13A2 affects the level of membrane-associated αsyn since polyamines electrostatically interact with biological membranes to bridge and shield the membrane surface charges that possibly contribute to the aggregation of membrane particles resistant to the repulsive forces [41]. Membrane fractionation experiments revealed that overexpression of ATP13A2 WT reduces αsyn levels at the membrane (Figure 3a,b), while KD of ATP13A2 induced a modest, nonsignificant increase in membrane-associated αsyn multimers (Figure 3c,d). We found similar effects in the presence of oxidative stress (Figure 3c,d). To investigate whether increased levels of αsyn at the membrane result in increased αsyn multimerization, we performed intact cell cross-linking before membrane isolation and assessed the cross-linked αsyn multimers in the membrane fraction. In line with the decreased level of membrane-associated αsyn, we could observe a reduction of membrane-associated αsyn multimers in ATP13A2 WT cells under basal conditions and during oxidative stress (Figure 3e,f). Interestingly, although oxidative stress did not increase the level of membrane-associated αsyn, it promoted the multimerization of membrane-associated αsyn in ATP13A2 KD, DN and fLuc cells. As in whole-cell extracts, this oxidative stress-induced multimerization at the membrane could also be prevented by ATP13A2 WT overexpression.

### 2.4. ATP13A2 Inhibits αsyn Multimerization via Regulating the UPS

We then wondered whether the observed effect of ATP13A2 WT on αsyn multimers was mediated by increased degradation. Therefore, we treated our cells with 10 µM of the lysosomal inhibitor chloroquine (CQ) for 24 h and found that autophagy inhibition did not affect the level of αsyn multimers in ATP13A2 WT cells compared to control conditions (Figure A3a–c, Appendix A). We also explored the involvement of chaperones in the clearance of αsyn multimers. We checked the expression levels of Hsc70 and Hsp90 but did not detect significant alterations in conditions with or without oxidative stress in any of the cell lines tested (Figure A3d,e, Appendix A). Next, we investigated the potential role of the UPS, given the previously reported impact of ATP13A2 on polyubiquitination [42]. Oxidative stress induced a significant increase in overall polyubiquitin levels, which was normalized by overexpression of ATP13A2 WT and remarkably ATP13A2 DN (Figure 4a,b). To determine whether this reduced polyubiquitination was related to the clearance of αsyn, we treated cells with 1 µM of the UPS inhibitor MG132 for 24 h and measured the levels of αsyn multimers. Inhibition of the UPS did not affect αsyn multimerization in basal conditions (Figure A3f, Appendix A). However, during oxidative stress, the protective role of ATP13A2 against αsyn multimerization was completely abolished upon UPS inhibition (Figure 4c,d or Figure A3g, Appendix A), pointing to an important role for proteasomal degradation of αsyn (multimers) in the observed effect of ATP13A2.

### 2.5. ATP13A2 Regulates SPM-Induced αsyn Multimerization

We recently showed that ATP13A2 is a lysosomal polyamine exporter with the highest affinity for SPM. At high concentrations, polyamines induce cell toxicity, which is exacerbated by ATP13A2 loss due to lysosomal dysfunction and rupture [32]. It was previously reported that SPM can promote αsyn aggregation in vitro [43]. Therefore, we investigated whether ATP13A2 influences αsyn multimerization in the presence of SPM. We observed that treatment with 5 µM SPM for 24 h (the concentration was chosen after optimization, considering its toxic effect and the viability of treated cells) induced a twofold increase in αsyn multimers in the control fLuc-overexpressing cells. ATP13A2 KD did not significantly enhance αsyn multimerization induced by SPM, while overexpression of ATP13A2 WT totally prevented this effect. The ATP13A2 DN cells were similar to the control cells, showing that the protective effect of ATP13A2 WT on SPM-induced αsyn multimerization is related to its transport function (Figure 5a,b or Figure A4, Appendix A).

Interestingly, while treatment with SPM did not affect αsyn levels at the membrane (Figure 5c,d), it resulted in significantly more membrane-bound αsyn in the membrane fraction in ATP13A2 KD cells (Figure 5e,f). When considering membrane-associated αsyn multimers, SPM challenge further enhanced the increase in ATP13A2 KD cells, which was again prevented by ATP13A2 WT overexpression but not DN (Figure 5e,f).

### 2.6. Impaired Lysosomal Membrane Integrity Is Linked to αsyn Multimerization

We previously reported that high concentrations of SPM induce the loss of lysosomal membrane integrity in ATP13A2 knockout (KO) cells [32]. Here, we aimed to link these findings to αsyn multimerization. To this end, we used acridine orange to detect the integrity of the lysosomes. Acridine orange is a weak base dye that becomes protonated under acidic conditions and gets trapped in lysosomes. However, if the lysosomal membrane is damaged, it can diffuse to the cytoplasm, where it gets deprotonated and shifts its emission from red to green fluorescence. We first compared the different αsyn expressing cell lines and found that the lysosomal membrane integrity was impaired in ATP13A2 KD cells in basal conditions, while overexpression of ATP13A2 WT or DN had no effect (Figure 6a), which is in line with previous findings [32]. We also validated the acridine orange assay by exposure of the cell lines to high concentration of chloroquine (500 µM, 2 h), which induces acute impairment of the lysosomal membrane (Figure 6b). Next, we observed that SPM treatment resulted in a greater loss of lysosomal membrane integrity only when ATP13A2 levels were reduced (Figure 6c). Therefore, we independently confirmed the essential role for ATP13A2 in maintaining the integrity of the lysosomal membrane during SPM-induced stress in a new cell model.

Of note, high CQ concentrations also induced a pronounced increase of αsyn multimers independent of ATP13A2 (Figure 6d,e or Figure A5, Appendix A), while low concentrations of CQ had no effect on αsyn multimerization both in basal and oxidative stress condition (Figure A3b,c, Appendix A). Thereafter, we studied the integrity of lysosomal membrane following the challenge with 10 µM CQ for 24h and confirmed that low concentrations of CQ are not sufficient to induce the damage of lysosomal membranes (Figure 6f), demonstrating that stress-induced lysosomal membrane damage may be linked to αsyn multimerization.

### 2.7. ATP13A2 Promotes Secretion of αsyn Multimers

Spreading of aggregated αsyn is considered as a key event in PD progression. Since ATP13A2 was reported to regulate the biogenesis of exosomes and promote αsyn secretion [15,27], we aimed to determine whether this effect relies on the ATPase function of ATP13A2. Therefore, we set out to investigate the level of extracellular αsyn carried by nanovesicles derived from the plasma membrane in our different cell models. By loading the same concentration of nanovesicle proteins, we found that overexpression of ATP13A2 WT and transport-deficient mutant ATP13A2 DN significantly increases the level of αsyn in nanovesicles (Figure 7a,b), suggesting that αsyn externalization is regulated by ATP13A2 via transport independent roles. Next, we performed cross-linking on the harvested intact nanovesicles to assess the level of αsyn multimerization in the secreted nanovesicles. We found that secretion of αsyn multimers was promoted by ATP13A2 WT and DN, while ATP13A2 KD did not have any significant effect. Treatment with SPM did not affect the secretion of αsyn multimers in any of the cell lines in line with the transport-independent effect of ATP13A2 on nanovesicle secretion, further indicating that nanovesicle secretion is not affected by lysosomal dysfunction (Figure 7c,d). Collectively, these data suggest that ATP13A2 can promote extracellular secretion of αsyn multimers through a transport-independent mechanism.

## 3. Discussion

In this study, we investigated in detail the relationship between ATP13A2 and αsyn multimerization, membrane localization, and secretion in basal conditions compared to conditions of oxidative stress and lysosomal rupture. Despite several previous studies having already considered the potential link between ATP13A2 and αsyn, such as those showing that deficiency of ATP13A2 leads to αsyn accumulation, as well as the protective effect of ATP13A2 against αsyn-induced neurotoxicity [21,22,28], the exact mechanisms underlying these effects remain incompletely resolved.

Our recent identification of SPM as a substrate of ATP13A2 sheds new light on the link between ATP13A2 and αsyn [32]. Polyamines enter the cells via endocytosis and, therefore, arrive in the lysosome. Since ATP13A2 functions as an SPM exporter in the lysosomal membrane, loss of ATP13A2 results in increased SPM trapped in the lysosomes, which leads to an overall lower polyamine content in cells. The accumulation of polyamines in the lysosome can ultimately causes lysosomal dysfunction and membrane rupture. We show that SPM exposure reduces lysosomal membrane integrity in our ATP13A2-deficient αsyn-expressing cells. Thus, lysosomal dysfunction and loss of lysosomal membrane integrity may, therefore, at least be partially responsible for the increased αsyn multimerization in ATP13A2-deficient cells.

We also found that deficiency in ATP13A2 increases αsyn multimerization in basal conditions. The levels of αsyn multimers induced by oxidative stress or SPM were increased in conditions of ATP13A2 KD and could be completely prevented upon overexpression of ATP13A2 WT, but not by the transport-deficient mutant DN. This is an interesting finding given the high oxidative stress levels in PD brain and highlights the importance of ATP13A2 in the prevention of αsyn multimerization in conditions of oxidative stress.

Under physiological conditions, αsyn populates an ensemble of confirmations by interacting with proteins or binding to membranes, mediating a variety of cellular events [9], whereas membrane-associated αsyn can also disrupt the lipid bilayer and induce pore-like structures, thereby accelerating αsyn aggregation [10,44,45]. Here, we found that reduced ATP13A2 levels result in increased levels of αsyn multimers at the membrane in conditions of oxidative stress or upon SPM treatment. Conversely, ATP13A2 WT prevents membrane αsyn multimerization in these stress conditions. Thus, ATP13A2 may regulate the levels of αsyn associated with the membrane, thereby controlling αsyn multimerization.

KD or KO of ATP13A2 may result in impaired autophagy, which has been linked to αsyn accumulation [21,46]. However, here, we first observed that increased overall polyubiquitin levels can be normalized by ATP13A2 WT and DN, suggesting that ATP13A2 can reduce polyubiquitination or promote degradation of polyubiquitinated proteins in a transport-independent manner, which is in line with previous findings [42]. Next, we found that inhibition of the ubiquitin-proteasome system but not the autophagy lysosomal pathway blocked the protective role of ATP13A2 against αsyn multimerization. This might indicate that ATP13A2-mediated removal of αsyn multimers occurs through the UPS in the conditions tested.

Spreading of pathological αsyn contributes to the neurodegenerative phenotype in a number of cell and animal models. Previous studies suggested the involvement of ATP13A2 in the biogenesis of exosomes and αsyn externalization [15,27], but the role of the transport function of ATP13A2 was not addressed. For this phenotype, we found that, independent of the transport function, ATP13A2 significantly increases the level of αsyn carried by nanovesicles, while ATP13A2 KD did not have any significant effect. In addition, by loading the same volume of nanovesicle samples from different cells, we also confirmed that ATP13A2 WT and DN can increase the biogenesis of exosomes, and that ATP13A2 KD seems to correlate with decreased level of exosome markers, which is in line with previous findings (Figure A6) [27]. These are interesting findings suggesting that ATP13A2 might be able to promote the sequestration of αsyn, which is subsequently externalized through nanovesicles. Moreover, when examined which αsyn species are externalized, strikingly, we found that, independent of the transport function, ATP13A2 significantly increases the release of αsyn multimers via nanovesicles. This is an intriguing finding, since one could argue that increased release of (multimeric) αsyn reduces the cellular αsyn levels but might at the same time promote the spreading of extracellular αsyn, thus amplifying the progression of the toxic effect from neurons to surrounding tissues. Thus, further studies will be required to determine whether these αsyn multimers in the nanovesicles are related to pathological spread in vivo or rather to a way for the cell to reduce its levels of αsyn (multimers) without detrimental effects on neighboring cells.

We used the artificial, transport-deficient mutant ATP13A2 DN as a control in several of our experiments, which enabled us to dissect polyamine transport-dependent and -independent functions of ATP13A2 [32]. ATP13A2 DN did not show significant effects in most conditions tested; however, it could prevent increased polyubiquitination induced via oxidative stress, as well as promote the externalization of αsyn multimers. Taken together, these findings suggest a scaffolding function of ATP13A2 next to the transporter role, which is in accordance with previous reports [42].

## 4. Materials and Methods

### 4.1. Generation of Stable Overexpression Cell Lines

Human neuroblastoma SH-SY5Y cells with stable expression of human αsyn WT were generated using pCHMWS-αsyn-IRES-puro lentiviral vectors (LVs), and cells were selected with puromycin (3 µg/mL, Gibco, Gaithersburg, MD, USA). Next, these selected cells were transduced with human ATP13A2-WT-IRES-puromycin, human ATP13A2-D508N-IRES-puromycin, fLuc-IRES-puromycin (control), sh-ATP13A2-blasticidin (KD), and sh-fLuc-blasticidin (KD control) LVs to obtain ATP13A2 overexpression or KD cells. All LVs were generated by the Leuven Viral Vector Core (LVVC) as described previously [42,47]. Cells were grown as monolayers and maintained in Dulbecco’s modified Eagle’s medium (DMEM) with Glutamax (Gibco, Gaithersburg, MD, USA), supplemented with 15% fetal calf serum (FCS; Gibco, Gaithersburg, MD, USA), 1% nonessential amino acids (NEAAs; Gibco, Gaithersburg, MD, USA), and gentamicin (50 μg/mL; Gibco, Gaithersburg, MD, USA) at 37 °C in a humidified atmosphere containing 5% CO_2_. After lentiviral transduction, cells were selected with blasticidin (9 µg/mL, Thermo Fisher Scientific, Waltham, MA, USA) or puromycin (3 µg/mL). To inhibit proteasomal or lysosomal activity, cells were treated for 24 h with 1 μM MG132 (Sigma-Aldrich, Saint Louis, MO, USA) or 10 μM chloroquine (CQ, Sigma-Aldrich, Saint Louis, MO, USA), respectively. For cell lysis, cells were washed with phosphate-buffered saline (PBS) and lysed in lysis buffer (Tris 20 mM pH 7.5, NaCl 150 mM, ethylenediaminetetraacetic acid (EDTA) 1 mM, Triton 1%, glycerol 10%, and protease inhibitor (PI) cocktail (Roche, Basel, Switzerland)). Cell lysates were cleared by centrifugation at 12,000× *g* for 15 min and further analyzed via immunoblotting.

### 4.2. Oxidative Stress Cell Model and SPM Addition

To induce αsyn aggregation, oxidative stress was applied to the cells as described by Macchi et al. with a few modifications [12]. One day after plating the cells in 10 cm dishes, cells were treated with 5 mM freshly prepared FeCl_2_ (Sigma-Aldrich, Saint Louis, MO, USA) and 100 µM H_2_O_2_ (Sigma-Aldrich, Saint Louis, MO, USA) in DMEM-complete (filtered through a 0.45µm filter (Merck Millipore, Burlington, MA, USA)). After 72 h, cells were washed with PBS and scraped for immunoblotting or intact cell cross-linking.

SPM (Sigma-Aldrich, Saint Louis, MO, USA) was prepared to a final stock concentration of 200 mM in 0.1 M 3-(N-morpholino)propane sulfonic acid (MOPS)-KOH (pH 7.0) [32]. For the SPM assay, cells were treated with 10 µM freshly prepared SPM for 24 h; then, cells were washed with PBS and scraped for intact cell cross-linking.

### 4.3. Intact Cell Cross-Linking

Cells were washed twice with PBS and harvested with PBS with PI cocktail (PBS + PI buffer). After centrifuging at 1000× *g* for 15 min, the cells were resuspended in 200 µL of PBS + PI buffer. Next, we incubated the cells with 0.5 μM DSG (Thermo Fisher Scientific, Waltham, MA, USA) or 1 μM DSP (Thermo Fisher Scientific, Waltham, MA, USA) for 30 min at 37 °C. The reaction was quenched with 10 µL of Tris-HCl buffer (1 M, pH 7.4) for 15 min at room temperature. Then cells were sonicated twice at 30 Hz for 15 on–off intervals of 10 s each (1 s pulse on/off) (Branson Sonifier, Ede, Netherlands).

### 4.4. Membrane Fractionation

Cells were washed twice with PBS, scraped, and harvested with PBS + PI buffer. After centrifuging at 1000× *g* for 15 min, the cells were resuspended in 500 µL of PBS + PI buffer and sonicated twice at 30 Hz for 15 on–off intervals of 10 s each (1 s pulse on/off). After sonication, cells were centrifuged at 340,000× *g* at 4 °C for 15 min, and the supernatant was collected as the cytosolic fraction. The pellets were washed three times with PBS + PI buffer and centrifuged at 340,000× *g* for 15 min; then, the pellets were dissolved in PBS + PI + 1% Triton buffer. After centrifugation at 340,000× *g* at 4°C for 15 min, the supernatant was collected as membranous fraction. For intact cell cross-linking combined with membrane fractionation experiments, cells were treated with cross-linker first, followed by membrane fractionation.

### 4.5. Nanovesicle Isolation

Cells were plated in 3 × 15 cm dishes and allowed to attach overnight. The next day, the cells were washed twice with PBS, and DMEM with 1% exosome-depleted fetal bovine serum (FBS) was added to each plate. The medium was collected 24 h later, and nanovesicles were purified using differential centrifugation: 300× *g* for 15 min to remove cells, 15,000× *g* for 30 min to remove cell debris, filtration of apoptotic bodies through a 0.22 µm filter, and a final spin at 140,000× *g* for 3 h at 4 °C. Supernatant was decanted, and nanovesicles were collected in 50 µL of cell lysis buffer for immunoblotting. For intact nanovesicle cross-linking, 9 × 15 cm dishes cells were plated. The nanovesicles were dissolved in 200 µL of PBS + PI buffer, followed by cross-linking with DSP or DSG.

### 4.6. Lysosomal Membrane Integrity

To assess lysosomal membrane integrity, cells were seeded in 12-well plates; the next day, cells were incubated with 5 µg/mL acridine orange (dissolved in media, Thermo Scientific, Waltham, MA, USA) for 15 min at 37 °C. Thereafter, medium was discarded, cells were washed with PBS, and fresh medium was added. Then, cells were treated with the indicated concentrations of SPM or the positive control chloroquine (500 µM) for 2 h at 37 °C. Lastly, cells were collected and resuspended in PBS containing 1% bovine serum albumin (BSA). The mean fluorescence of 10,000 events was captured using an Attune Nxt (Thermo Scientific, Waltham, MA, USA) flow cytometer.

### 4.7. Immunoblotting

After determination of protein content using a bicinchoninic acid (BCA) assay (Pierce Biotechnology, Waltham, MA, USA), cell extracts containing 15 µg of total protein were separated on 4–15% precast midi protein gels (Bio-Rad, Hercules, CA, USA) and electroblotted onto polyvinylidene difluoride membranes (Bio-Rad, Hercules, CA, USA). Membranes were first incubated with 0.4% paraformaldehyde (PFA) for 15 min for better αsyn binding, then blocked with PBST with 5% milk for 30 min and incubated with the following primary antibodies: anti–α-Syn (BD Transduction, San Jose, CA, USA), anti-α-tubulin (Sigma-Aldrich, Saint Louis, MO, USA), anti-ubiquitin (Abcam, Cambridge, MA, USA), anti-GAPDH (Abcam, Cambridge, MA, USA), anti-β-catenin (Abcam, Cambridge, MA, USA), anti-Hsc70 (Abcam, Cambridge, MA, USA), anti-Hsp90 (Abcam, Cambridge, MA, USA), anti-LC3 (Novus Biologicals, Centennial, CO, USA), and anti-ATP13A2 (Sigma-Aldrich, Saint Louis, MO, USA). After overnight incubation with primary antibodies at 4 °C, blots were washed three times with PBS/1%Triton and incubated with horseradish peroxidase-conjugated secondary antibodies (Dako, Glostrup, Denmark) for 1 h. Detection was performed using chemiluminescence (Thermo Fisher Scientific, Waltham, MA, USA) and an LAS-4000 imaging system (Fujifilm, Saitama, Japan).

### 4.8. Statistics

Figures shown are representative of at least three independent experiments. Images were analyzed by ImageJ 1.49v, and data were analyzed using Graph Pad Prism 6 (version 6.00, GraphPad Software, San Diego, CA, USA). Statistical analysis was performed using a one-way ANOVA test with Bonferroni to compare selected pairs of columns or column statistics (one-sample *t*-test) to compare values to the hypothetical value of 1. All error bars represent the standard error of the mean (SEM). Statistical significance was set at *** *p* < 0.001, ** *p* < 0.01, and * *p* < 0.05.

## 5. Conclusions

Our findings indicate that ATP13A2 regulates αsyn multimerization induced by oxidative stress or SPM via a mechanism involving maintenance of lysosomal membrane integrity, regulation of αsyn membrane association, and UPS-mediated αsyn degradation. Our findings shed new light on the link between αsyn and ATP13A2 and provide an explanation for the protective role of ATP13A2.

## Figures and Tables

**Figure 1 ijms-22-02689-f001:**
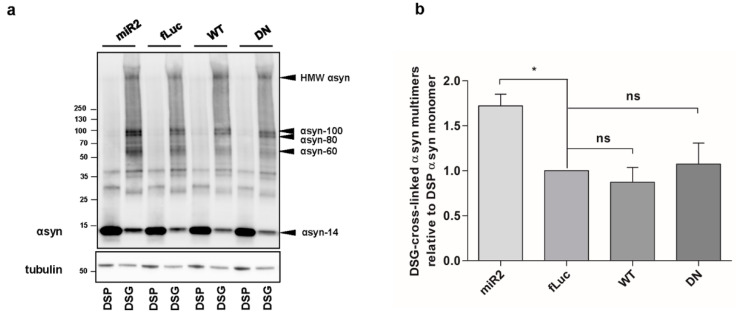
ATP13A2 knockdown (KD) results in upregulation of αsyn multimers in SH-SY5Y cells. (**a**) Visualization of α-synuclein (αsyn) multimers by Western blot after cross-linking in ATP13A2 KD, control fLuc, ATP13A2 wildtype (WT), and ATP13A2 D508N (DN) cells. (**b**) Quantification of disuccinimidyl glutarate (DSG) cross-linked αsyn-60 + αsyn-80 + αsyn-100 + HMW (high molecular weight) αsyn relative to dithiobis succinimidyl propionate (DSP) αsyn-14 monomer in the ATP13A2 KD, ATP13A2 WT, and ATP13A2 DN cells relative to the ratio in the control cells (*n* = 4). Statistical analysis was performed using a one-way ANOVA test with Bonferroni. The error bars represent the standard error of the mean (SEM). * *p* < 0.05, ns, not significant. αsyn-60 = ~60 kDa αsyn, αsyn-80 = ~80 kDa αsyn, αsyn-100 = ~100 kDa αsyn.

**Figure 2 ijms-22-02689-f002:**
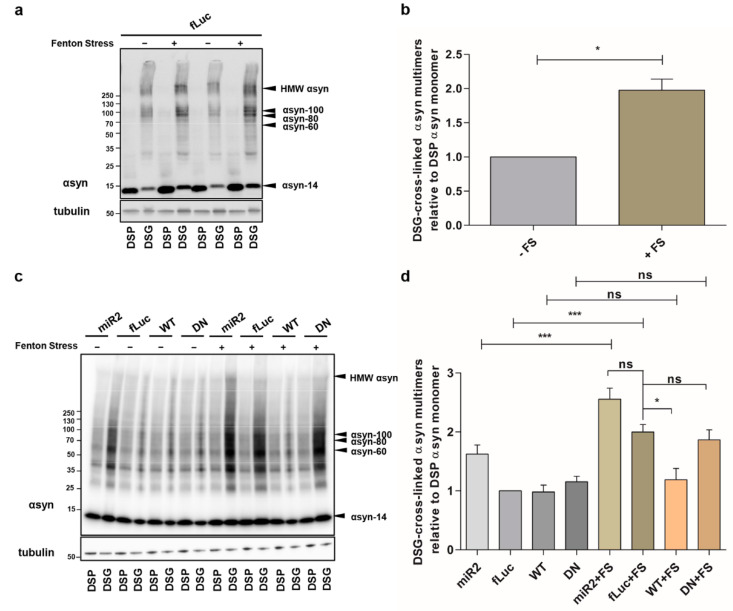
ATP13A2 regulates oxidative stress-induced αsyn multimerization. (**a**) Visualization of αsyn multimers after cross-linking in control cells without and with oxidative stress. (**b**) Quantification of the αsyn multimer-to-monomer ratio relative to the ratio in the control cells (*n* = 3). Statistical analysis was performed with column statistics (one-sample *t*-test) comparing test values to the hypothetical value of 1. The error bar represents the SEM. * *p* < 0.05. (**c**) Visualization of αsyn multimers after cross-linking in ATP13A2 KD, fLuc, ATP13A2 WT, and ATP13A2 DN cells without and with oxidative stress. (**d**) Quantification of the αsyn multimer-to-monomer ratio relative to the ratio in the control fLuc cells (*n* = 3). Statistical analysis was performed using a one-way ANOVA test with Bonferroni. The error bars represent the SEM. *** *p* < 0.001, * *p* < 0.05, ns, not significant.

**Figure 3 ijms-22-02689-f003:**
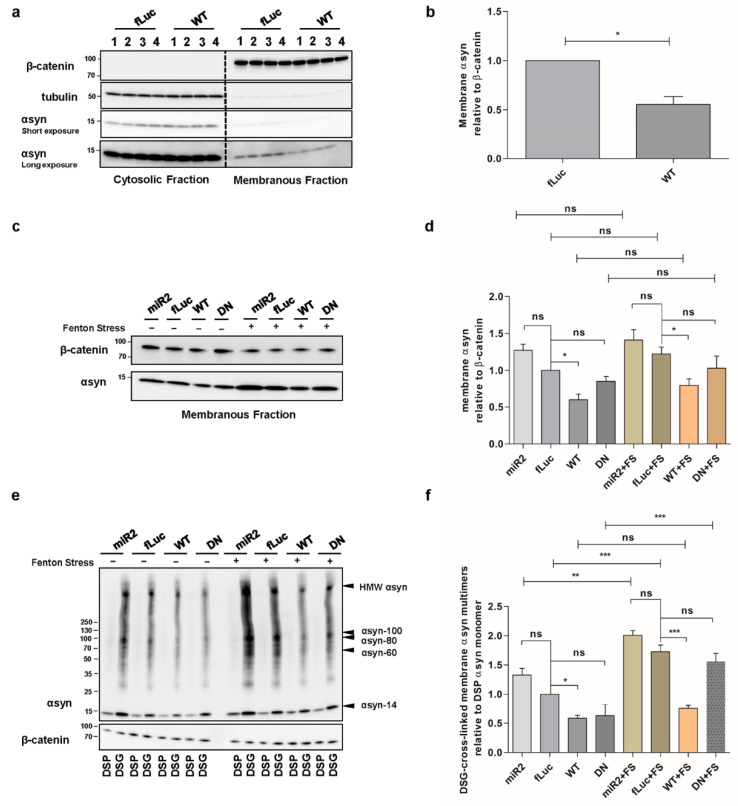
ATP13A2 regulates αsyn membrane association and multimerization. (**a**) Analysis of αsyn protein levels by Western blot in the cytosolic and membrane fraction of control and ATP13A2 WT cells using β-catenin as a marker for the membrane fraction and tubulin as a marker for the cytosolic fraction. (**b**) Quantification of membrane-associated αsyn relative to β-catenin (*n* = 3). Statistical analysis was performed with column statistics (one-sample *t*-test) comparing test values to the hypothetical value of 1. The error bars represent the SEM. * *p* < 0.05. (**c**) Analysis of αsyn protein levels in the membrane fraction of ATP13A2 KD, control, ATP13A2 WT, and ATP13A2 DN cells without and with oxidative stress by Western blot using β-catenin as a standard. (**d**) Quantification of membrane-associated αsyn relative to β-catenin (*n* = 3). Statistical analysis was performed using a one-way ANOVA test with Bonferroni. The error bars represent the SEM. * *p* < 0.05, ns, not significant. (**e**) Cell lysates of cross-linked membrane fractions from ATP13A2 KD, control, ATP13A2 WT, and ATP13A2 DN cells without and with oxidative stress. (**f**) Quantification of ratio of membrane-associated αsyn multimers to DSP-αsyn. Statistical analysis was performed using a one-way ANOVA test with Bonferroni. The error bars represent the SEM. *** *p* < 0.001, ** *p* < 0.01, * *p* < 0.05, ns, not significant.

**Figure 4 ijms-22-02689-f004:**
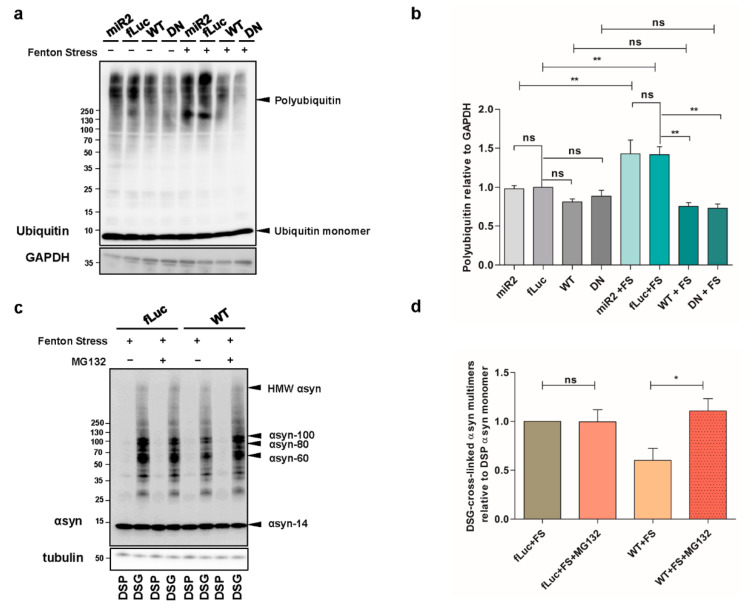
ATP13A2 inhibits αsyn multimerization via regulating the ubiquitin-proteasome system (UPS). (**a**) Visualization of ubiquitin levels in cell lysates from ATP13A2 KD, fLuc, ATP13A2 WT, and ATP13A2 DN cells without and with oxidative stress treatment. Glyceraldehyde 3-phosphate dehydrogenase (GAPDH) levels were used as standard. (**b**) Quantification of polyubiquitin levels relative to glyceraldehyde 3-phosphate dehydrogenase (GAPDH) (*n* = 4). Statistical analysis was performed using a one-way ANOVA test with Bonferroni. The error bars represent the SEM. ** *p* < 0.01, ns, not significant. (**c**) Visualization of αsyn multimers after cross-linking in fLuc and ATP13A2 WT cells under oxidative stress without and with 1 µM MG132. (**d**) Quantification of αsyn multimer-to-monomer ratio (*n* = 4). Statistical analysis was performed with column statistics (one-sample *t*-test) comparing test values to the hypothetical value of 1. * *p* < 0.05, ns, not significant.

**Figure 5 ijms-22-02689-f005:**
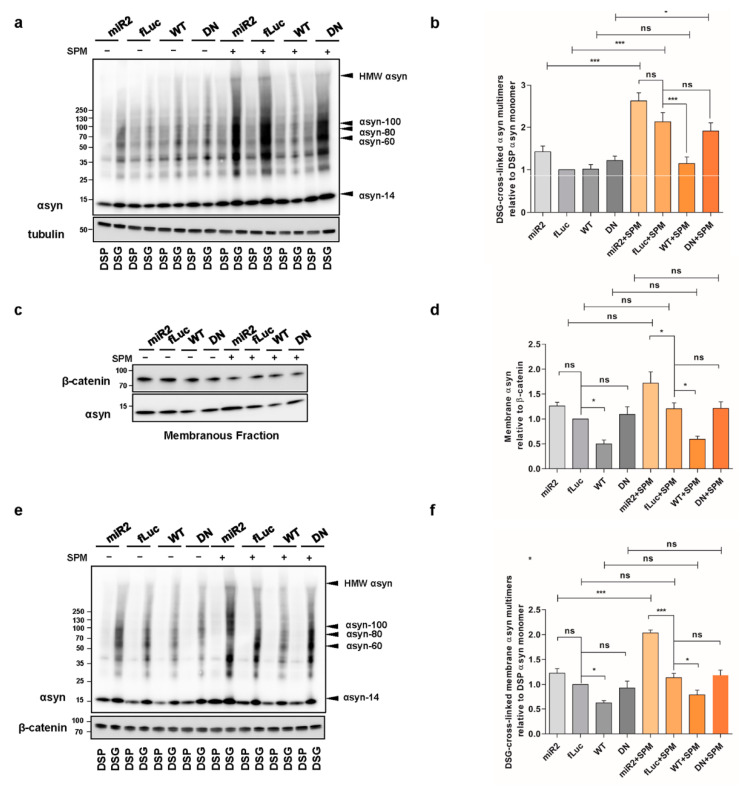
ATP13A2 regulates SPM-induced αsyn multimerization. (**a**) Visualization of αsyn multimers after cross-linking in ATP13A2 KD, control, ATP13A2 WT, and ATP13A2 DN cells without and with SPM treatment. (**b**) Quantification of the αsyn multimer-to-monomer ratio relative to the ratio in the control cells (*n* = 3). Statistical analysis was performed using a one-way ANOVA test with Bonferroni. The error bar represents the SEM. *** *p* < 0.001, * *p* < 0.05, ns, not significant. (**c**) Visualization of αsyn protein levels in the membrane fraction of ATP13A2 KD, control, ATP13A2 WT, and ATP13A2 DN cells without and with SPM treatment were analyzed by Western blot using β-catenin as a standard. (**d**) Quantification of membrane-associated αsyn relative to β-catenin (*n* = 3). Statistical analysis was performed using a one-way ANOVA test with Bonferroni. The error bars represent the SEM. * *p* < 0.05, ns, not significant. (**e**) Visualization of αsyn multimers after cross-linking in membrane fraction from ATP13A2 KD, fLuc, ATP13A2 WT, and ATP13A2 DN cells without and with oxidative stress. (**f**) Quantification of ratio of membrane-associated αsyn multimers to DSP-αsyn monomer (*n* = 3). Statistical analysis was performed using a one-way ANOVA test with Bonferroni. The error bars represent the SEM. *** *p* < 0.001, * *p* < 0.05, ns, not significant.

**Figure 6 ijms-22-02689-f006:**
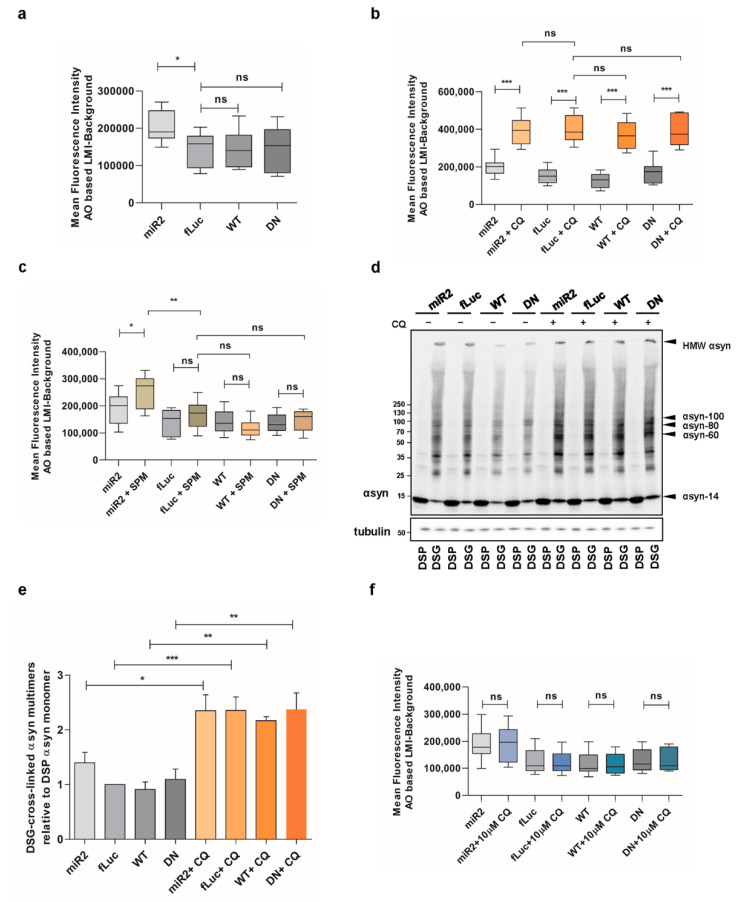
Impaired lysosomal membrane integrity is linked to αsyn multimerization. (**a**) Assessment of lysosomal membrane integrity via acridine orange staining in basal conditions (*n* = 9). Statistical analysis was performed using a one-way ANOVA test with Bonferroni. The error bars represent the SEM. * *p* < 0.05, ns, not significant. AO = acridine orange, LMI = lysosomal membrane integrity. (**b**) Assessment of lysosomal membrane integrity via acridine orange staining in ATP13A2 KD, fLuc, ATP13A2 WT, and ATP13A2 DN cells without and with chloroquine (CQ) (*n* = 9). Statistical analysis was performed using a one-way ANOVA test with Bonferroni. The error bars represent the SEM. *** *p* < 0.001, ns, not significant. (**c**) Assessment of lysosomal membrane integrity via acridine orange staining in ATP13A2 KD, fLuc, ATP13A2 WT, and ATP13A2 DN cells without and with SPM (*n* = 9). Statistical analysis was performed using a one-way ANOVA test with Bonferroni. The error bars represent the SEM. ** *p* < 0.01, * *p* < 0.05, ns, not significant. (**d**) Visualization of αsyn multimers after cross-linking in control cells without and with a high concentration of CQ. (**e**) Quantification of the αsyn multimer-to-monomer ratio relative to the ratio in the control cells (*n* = 3). Statistical analysis was performed using a one-way ANOVA test with Bonferroni. The error bars represent the SEM. *** *p* < 0.001, ** *p* < 0.01, * *p* < 0.05. (**f**) Assessment of lysosomal membrane integrity via acridine orange staining in ATP13A2 KD, fLuc, ATP13A2 WT, and ATP13A2 DN cells without and with a low concentration of CQ (*n* = 9). Statistical analysis was performed using a one-way ANOVA test with Bonferroni. The error bars represent the SEM. ns, not significant.

**Figure 7 ijms-22-02689-f007:**
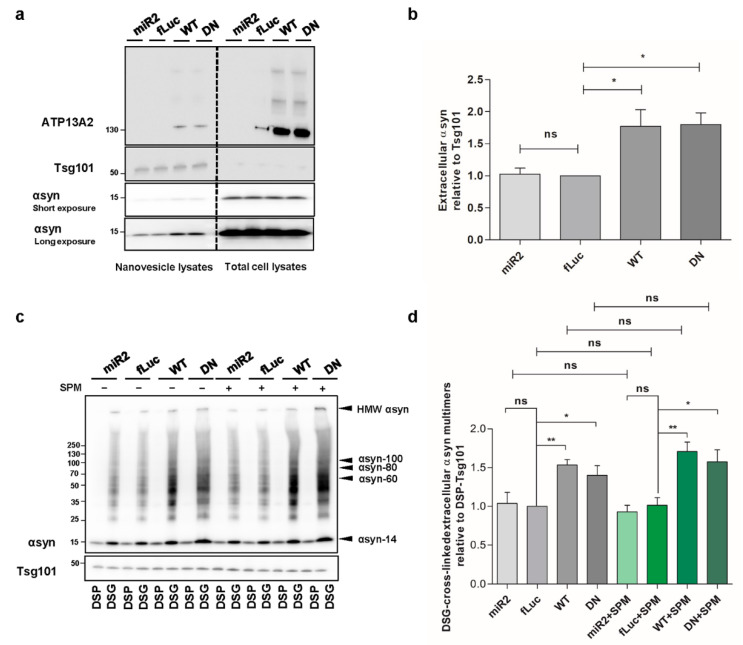
ATP13A2 promotes secretion of αsyn multimers. (**a**) Visualization of αsyn protein levels in total cell lysates and nanovesicle extracts from ATP13A2 KD, control, ATP13A2 WT, and ATP13A2 DN cells using Tsg101 as a control for the nanovesicle fraction. (**b**) Quantification of αsyn relative to Tsg101 (*n* = 3). Statistical analysis was performed using a one-way ANOVA test with Bonferroni. The error bars represent the SEM. * *p* < 0.05, ns, not significant. (**c**) Visualization of αsyn multimers after cross-linking in nanovesicles from ATP13A2 KD, control, and ATP13A2 DN cells without and with SPM using Tsg101 as a control for the nanovesicle fraction. (**d**) Quantification of ratio of αsyn multimers to DSP-Tsg101. Statistical analysis was performed using a one-way ANOVA test with Bonferroni. The error bars represent the SEM. ** *p* < 0.01, * *p* < 0.05, ns, not significant.

## Data Availability

All data are reported in the manuscript and in the Appendix A.

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
