# Peer review of "ATP13A2 Regulates Cellular α-Synuclein Multimerization, Membrane Association, and Externalization"

_ijms, 2021, doi:10.3390/ijms22052689_

Round 1

Reviewer 1 Report

Manuscript ID: ijms-1113195

In their article “ATP13A2 regulates cellular α-synuclein multimerization, membrane association and externalization“, Jianmin Si and colleagues present an interesting study on the interplay of ATP13A2 and αsyn, two proteins involved in neurodegenerative diseeases. By using SHSY5Y neuroblastoma cell and knockdown as well as overexpression of ATP13A2 and respective controls, the authors analyzed the role of ATP13A2 in the pathophysiology of αsyn. Using crosslinking technology it was shown that knockdown of ATP13A2 increases αsyn multimers, which was even further increases under oxidative stress conditions, and here especially for membrane-associated αsyn. The authors link this finding to an ATP13A2-mediated regulation of the ubiquitin-proteasome system and an influence of the polyamine spermine. Finally, the authors show that ATP13A2 affects lysosomal membrane integrity and promotes the secretion of αsyn via nanovesicles.

  • p. 2, line 58: the use of “cause” in this sentence is too strong, since there are multiple factors involved in the development of neurodegenerative diseases. I suggest to use an expression like “Different mutations of ATP13A2 have been shown to be involved in…”
  • p. 6, line 197ff: this sentence should be moved to the discussion part.
  • p. 9, line 260: same as above.
  • p. 10, Fig. 6: Could the authors provide exemplary pictures of the SHSY5Y cells or a scatter plot from the flow cytrometry analysis? Please explain abbreviations (CQ, AO, LMI) in the figure legend.
  • p.12, line 323: “such us” should read “such as”
  • p. 13, M&M section: the authors constantly use the term “2 times”, which should be changed to “twice”

Reviewer 2 Report

This manuscript sough to identify mechanism(s) regarding the involvement of ATP13A2 in regulating the fate of alpha-synuclein including cytoplasmic accumulation and external release. The abstract states that loss of ATP13A2 impairs lysosomal membrane integrity and induces α-synuclein multi-merization at the membrane. In contrast, overexpression of ATP13A2 WT had a protective effect on α-synuclein multimerization, which corresponded with reduced αsyn membrane association. Lastly that ATP13A2 (mutant or WT) promoted the secretion of α-synuclein through nanovesicles.

General Comment:  The authors sought “to explore the initial stages of αsyn aggregation by investigating αsyn multimerization” but no data is presented that these cross-linked aSyn multimers (αsyn-60 + αsyn-80 + αsyn-100) detected by immunoblot :

- represent the initial stages of aSyn aggregation. ie: that these aSyn multimers are on the pathway towards becoming aggregates.  It is surprising that intacellular aSyn aggregates are never examined in the manuscript, despite the authors having published a ThT-based assay to measure aSyn aggregates.

- whether the described “multimerizated aSyn species” are toxic or not

- are the “multimerizated aSyn species” relevant to aSyn pathology and the disease process of Parkinson's.

The authors need to demonstrate the relevance of the cross-linked multimerizated aSyn species (αsyn-60 + αsyn-80 + αsyn-100). Without providing evidence that the described “multimerizated aSyn species” are relevant to intracellular aSyn aggregate formation, cellular toxicity and/or the pathophysiology of aSyn, then the direct relevance to PD of the submitted findings  is unknown. Unfortunately, all of the remaining experiments (figs 3,4,5,6,7) utilise this cross-linked aSyn multimerisation approach, and thus the relevance of the findings from these figures is also unclear.

Similarly, native non-pathogenic aSyn associates with membranes that include synaptic vesicles but the manuscript conveys that "membrane-bound aSyn was reported to have a higher propensity to aggregate", leaving the implications unclear when changes in aSyn membrane association were observed. The authors should test/confirm if alterations of aSyn membrane association in their experimental system correlates with aSyn aggregation.

Specific comments on the section: “ATP13A2 promotes secretion of αsyn multimers”

The authors collected nanovesicles but co-sedimentation of Tsg101 and aSyn from a 15,000g supernatant by a 3 hour 140,000g spin is not evidence that nanovesicles have been purified, nor that aSyn is internal to the nanovesicles, just that that centrifugation protocol pelleted both of these proteins. Substantially more evidence is needed to demonstrate that the authors have purified nanovesicles, and that aSyn is within the lumen of the nanovesicles. Having accomplished this, is would be beneficial if the authors were to incubate the nanovesicles with untreated cells to demonstrate the spreading of αsyn pathology.

The author’s observation that knockdown of ATP13A2 in this system caused no reduction in externalised aSyn contradicts the published results in ref #26 “ATP13A2/PARK9 regulates secretion of exosomes and alpha-synuclein”, and should be discussed/addressed.

Reviewer 3 Report

This manuscript describes a carefully performed study on the characteristics of the polyamine transporter ATP13A2 and its effects on alpha-synuclein aggregation, membrane binding and cellular export.  Each of the experiments demonstrate a relationship between the presence/absence of ATP13A2 and a specific aspect of alpha synuclein cellular dynamics.  Although the paper does not provide us with specific mechanisms regarding each of these relationships, it serves to provide various pointers for future studies that will address them in future.  

Author Response

We would like to express our deepest appreciation for your interest and recognition of our work!